# Is the Prevalence of *Leishmania infantum* Linked to Breeds in Dogs? Characterization of Seropositive Dogs in Ibiza

**DOI:** 10.3390/ani11092579

**Published:** 2021-09-02

**Authors:** Maria Edo, Pablo Jesús Marín-García, Lola Llobat

**Affiliations:** 1Facultad de Veterinaria, Universidad Cardenal Herrera-CEU, CEU Universities, 46113 Valencia, Spain; maria.edo1@alumnos.uchceu.es; 2Department of Animal Production and Health, Veterinary Public Health and Food Science and Technology (PASAPTA), Facultad de Veterinaria, Universidad Cardenal Herrera-CEU, CEU Universities, 46113 Valencia, Spain

**Keywords:** *Leishmania infantum*, dog, Ibiza, prevalence, infection

## Abstract

**Simple Summary:**

Leishmaniosis is an important zoonotic protozoan disease. *Leishmania infantum* is a protozoan species that accounts for the majority of cases in the Mediterranean. In this study, we analyzed the prevalence of infection in different dog breeds from Ibizan Island. Our results showed that the Doberman Pinscher and Boxer breeds present a higher prevalence of infection, and the relationship between antibodies’ serum titer and staging of disease was confirmed. Differences between age and sex were not found.

**Abstract:**

Leishmaniosis is an important zoonotic protozoan disease primarily spread to the Mediterranean region by *Leishmania infantum*, the predominant protozoan species, which accounts for the majority of cases. Development of disease depends on the immune response of the definitive host and, predictably, their genetic background. Recent studies have revealed breed-typical haplotypes that are susceptible to the spread of the protozoan parasite. The objective of this study was to analyze the prevalence of leishmaniosis on a Mediterranean island and determine the relationship between disease prevalence and breed. In addition, information on seropositive animals was recorded to characterize animals affected by the disease. To study the prevalence, a total of 3141 dogs were analyzed. Of these, the 149 infected animals were examined for age, sex, antibody titer, and disease stage. We observed a prevalence of 4.74%, which varied between breeds (*p* < 0.05). The Doberman Pinscher and Boxer breeds had the highest prevalence of leishmaniosis. Significant differences were observed between breeds with common ancestors, emphasizing the important genetic component. Finally, regarding the characterization of seropositive animals, the distribution is similar to other studies. We discovered a relationship (*p* < 0.05) between the number of antibody titers and the clinical disease stage, which was also present in *Leishmania infantum*, suggesting that the development of the disease depends on the humoral or Th2 immune response with ineffective antibodies.

## 1. Introduction

Leishmaniosis is a parasitic disease caused by different genera of *Leishmania*, including *Leishmania infantum* (*L. infantum*), the most prevalent causal agent of leishmaniosis in the Mediterranean area stretching as far as the Iberian Peninsula. This zoonotic disease is endemic to 88 countries and is considered the most relevant vector-borne disease in the Mediterranean, affecting between 63% and 80% of the domestic dog population [1,2,3,4,5]. Akohundi et al., (2016) identified 53 different species of the *Leishmania* genera and more than 800 vectors capable of transmitting the infection stadium of the parasite (promastigote) to the host [6]. Although domestic dogs are the definitive hosts, different studies have documented *L. infantum* in other mammalian species, including horses, cats, pigs, wild rodents, hares, wild canids, and humans [7,8,9,10,11,12,13,14]. The host immunity response determines the infection severity of *L. infantum*, which can be mild, moderate, severe, or very severe [15]. Solano–Gallego et al. (2009) reported that the most common clinical signs of infection by *L. infantum* are skin lesions, generalized lymphadenomegaly, progressive weight loss, muscular atrophy, exercise intolerance, decreased appetite, lethargy, splenomegaly, polyuria and polydipsia, ocular lesions, epistaxis, onychogryphosis, lameness, vomiting, and diarrhea [15]. In terms of spatial distribution, the seroprevalence *of L. infantum* varies in different Spanish areas, ranging from 3.7% to 34.6% in the north and south of the country, respectively [16,17]. There are some studies on the prevalence of *L. infantum* on Mediterranean islands [18,19]. Burnham et al., (2020) studied the relationship between anti-*Phlebotomus pernicious* saliva antibodies in Ibizan hounds and susceptibility to canine leishmaniosis without analyzing seroprevalence data in different dog breeds from the Mediterranean coast of Spain [20]. The global seroprevalence in domestic dogs of the Spanish Mediterranean islands was estimated at approximately 57.1% [21]. Although numerous studies (in both visceral and cutaneous forms) explored a possible genetic influence or resistance to disease in murine and human models, similar studies in dogs are scarce [22,23,24,25,26,27,28]. However, some studies found resistance and/or susceptibility to disease depending on the dog breed, thereby influencing the pathological process [29,30]. Recently, Vasconcelos et al., (2019) published a review on dogs’ genetic resistance to infection by *Leishmania* spp., particularly the Ibizan Hound breed, but did not obtain clear evidence [31]. These data complement Solano-Gallego et al.’s study (2000), which demonstrated that the Ibizan Hound presents a greater immune response than dogs of other breeds [30].

The clinical pathogeny of *L. infantum* infection is a multifactorial process, where the activation of Th1 or Th2 pathways depends on the host–pathogen interaction. The cellular immune response or Th1 pathway eliminates amastigotes within the macrophage more efficiently since stimulation by cytokines, mainly interferon-gamma (IFN-γ) and interleukin-2 (IL-2), cause a de-inhibition of nitric oxide production and deactivate the action of the arginase enzyme. The relationship between nitric oxide and arginase is essential for the progression of *L. infantum* inside macrophages [32,33,34]. However, recent studies have linked clinical symptoms to an increase in the expression of transforming necrosis factor-alpha (TNF-α), IL-1β, IL-4, IL-10, and IL-12, suggesting that the balance between different cytokines plays an important role in regulating inflammation and clinical presentations of the disease [35,36]. Hosts who activate the humoral immune response or Th2 pathway develop the disease with greater prevalence and severity since the antibodies are ineffective and stimulation by cytokines is based on the production of IL-10. This situation favors the development of the parasite along with the arginase enzyme and synthesis of polyamides [37]. Studies in vitro have demonstrated that blocking the activity of receptor IL-10 may be a new effective treatment [38,39].

Some authors have associated this differential immune response to the genetic backgrounds of these canine breeds. Additionally, different canine breeds present different prevalence rates [40,41]. Genes related to the activation of Th1 pathways include *IL12RB1, JAK3*, *CCRL2*, *CCR2*, *CCR3*, and *CXCR6*, whereas *COMMD5* and *SHARPIN* are associated with the activation of Th2 pathways [37]. Other genes, such as *Slc11c11* or Major Histocompatibility Complex I gene (*MHCI*), also affect the susceptibility or resistance to *L. infantum* infection [31]. Batista et al., (2020) associated genetic markers with antibody response in infected dogs [42]. Furthermore, different dog breeds show differences in gene expression. Sánchez-Roberts et al., (2008) postulated that some breeds, such as the Boxer, are most susceptible to infection [43] due to gene expression. These authors conducted a study with 19 breeds (including the Ibizan Hound and Boxer), where they analyzed the presence of polymorphism in the *Slc11a1* gene, which is related to autoimmune diseases in humans [44]. They concluded that two of the 24 polymorphisms found in this gene showed greater susceptibility to infection. Six single nucleotide polymorphisms (SNPs) in three different genes were detected in the Beagle breed, correlating with susceptible infection phenotypes [29].

Despite several studies regarding the genetic resistance and susceptibility of canine breeds, few have determined the prevalence of this infection in different breeds. Therefore, we aimed to investigate the effect of breed on *L. infantum* prevalence and characterize the profile of seropositive animals on a Mediterranean island as well as to identify and characterize the profile of seropositive animals.

## 2. Materials and Methods

### 2.1. Ethics Approval

The experiments involving animals were conducted according to the guidelines of the Declaration of Helsinki and approved by the Animal Experimentation Ethics Committee of the Universidad Cardenal Herrera CEU, with code 2020/VSC/PEA/0216.

### 2.2. Data and Collection of Samples

The study was conducted on Ibiza (38°54′31.79″ N, 1°25′58.66″ E; an area of 572 km^2^), a Mediterranean island located east of Spain. Data and blood samples from 3141 dogs with symptoms of *L. infantum* infection were collected between September 2020 and May 2021 (Table 1) [2]. Blood samples were obtained in 5 mL EDTA tubes from all symptomatic animals by cephalic venipuncture for serological analysis. Serological analysis was performed for all animals, including those with a positive serological test. The data recovered included prevalence (2 levels), breed (25 levels), sex (2 levels), age (divided into 3 categories: young (<4 years), adult (4 ≤ years < 10), and older dogs (≥10 years)), presentation of disease (3 levels: cutaneous, visceral, or both), and clinical stage. Four levels (Table 1) were assigned to each animal in a physical exam evaluating clinical signs according to Solano-Gallego et al. [15].

### 2.3. Serologic Tests

Blood samples were collected from the cephalic vein in 5 mL EDTA tubes. Plasma was obtained and preserved at −20 °C before analysis. Serological testing for *L. infantum* detected specific antibodies using the indirect immunofluorescence antibody test (IFAT), which was conducted by an external laboratory, and the IFAT for anti-*Leishmania*-specific immunoglobulin G (IgG) antibodies (MegaFLUO LEISH^®^, Megacor Diagnostik GmbH, Hörbranz, Austria). Seroprevalence was calculated as the percentage of dogs testing positive for *L. infantum* antibodies. Dog serum was considered seropositive with IFAT titer ≥ 1:80, following the manufacturer’s instructions [45,46]. The titer of detected antibodies was classified into 3 levels: low or questionable (<1:100), medium (between 1:100 and 1:400), and elevated (>1:400).

### 2.4. Statistical Analysis

Seroprevalence status was analyzed using the GENMOD procedure of the statistical program SAS (North Carolina State University, USA). Within positive animals, antibody titers were analyzed using a binary (in the case of sex) or multivariable (in the case of breed, age, presentation of disease, and clinical stage) logistic regression model using each factor as a fixed effect in their respective statistical analysis. The statistical significance was set at *p*-value < 0.05.

## 3. Results

Table 2 shows the seroprevalence (%) observed in different breeds using the cut-off of an *L. infantum* antibody titer ≥1:100 to denote seropositivity. The total seroprevalence was 4.74% (149 to 3141 animals studied) and varied between the canine breeds studied (*p*-value < 0.05). The results showed three groups evaluated for seroprevalence: The first group (Table 2, superscript a) with the lowest prevalence (on average 1.73%) comprised the Pug (0.72%), French Bulldog (1.41%), Maltese Bichon (1.47%), Chihuahua (2.33%), and crossbreeds (2.74%). The second group (Table 2, superscript b), with medium prevalence, contained only the German Shepherd (7.38%). The third group (Table 2, superscript c) with the highest prevalence (34.26% on average) comprised the Pointer (24.49%), Dogue de Bordeaux (33.33%), Great Dane (33.33%), Majorcan Shepherd (33.33%), Weimaraner (33.33%), Boxer (39.13%), and Doberman Pinscher (42.86%) breeds. The animals that do not present statistically significant differences are indicated by more than one superscript. Dogs belonging to the Beagle breed (Table 2, with superscripts a and b) only show statistically significant differences compared with the Boxer breed (Table 2, superscript a) and German Shepherd (Table 2, superscript b). 

Table 3 describes the seropositive animals based on sex, age, presentation of the disease, and clinical status. The number of seropositive males was higher than females. Younger dogs and adults had higher seropositivity than older dogs, with a cutaneous presentation of the disease in clinical stage II (Table 3).

No observable effects were detected between antibody titers, and canine breed, sex, or age. However, antibody titers were recorded in seropositive animals ranging from 1:100 to 1:1280, which is associated with presentation of disease (*p*-value < 0.05) but not canine breed, sex, or age (*p*-values of 0.97, 0.46, and 0.43, respectively). 

Only one dog with a high level of antibodies (>1:400) presented as clinical stage I (*p*-value < 0.05). Animals with a low antibody titer (<1:100) had clinical stages III or IV (Figure 1).

## 4. Discussion

In this work, we observed a seroprevalence of *L. infantum* infection of 4.74%. This result agrees with data observed in other reported studies on Mediterranean areas, where the prevalence was estimated between 5% and 57.1% [16]. In the Balearic Islands, the prevalence was approximately 1% of 813 dogs in 1999 [47] and 40% of 40 dogs ranging from 2011 to 2016 [21]. These relevant differences may be due to the sample size used and asymptomatic animals. Our study and others, which demonstrated high prevalence, only analyzed animals with symptoms, suggesting that the actual prevalence may be higher. Our work is the first to analyze and document more than 3000 dogs from the island of Ibiza. In other Spanish regions such as Northern Spain, the prevalence rate is lower than in Mediterranean regions at approximately 4% [17,45], resembling our data. These differences are likely due to environmental conditions and the vector distribution. Gálvez et al., (2020) analyzed the seroprevalence of *L. infantum* in dogs and sand fly distribution in Spain. They confirmed that the Balearic Islands are hyperendemic with *P. perniciosus* as the most abundant sand fly vector [21]. On the one hand, the climate and vegetation of Northern Spain are typically oceanic (warm summers and cool winters) and distribute rain throughout the year. On the other hand, the Mediterranean climate is characterized by mild and rainy winters, and dry and hot summers, with variable autumn and spring seasons (both in temperature and rain). This situation promotes the dispersion of *L. infantum* transmitted vectors [48]. Although the sample sizes were different between the breeds, the robustness of our statistical analysis indicates that differences exist. However, other authors conducted similar studies with similar sample sizes, and the results obtained are comparable to those of our study [20,49,50]. Regarding canine breeds, the high seroprevalence found in Boxers and Doberman Pinschers, 39.13% and 42.86%, respectively, agree with other study results [2,41,49,51], where authors attributed the genetic susceptibility and high prevalence of the Boxer breed to the *TAG-8-141* haplotype. Abranches et al., (1991) conducted a study in Portugal showing high prevalence in the Doberman Pinscher and German Shepherd breeds [51]. However, the latter breed had a moderate seroprevalence in our study. Other authors suggested that the elevated susceptibility of the Boxer and Doberman Pinscher is due to three single nucleotide polymorphisms (SNPs) in the *Slc11a1* gene, specifically in the promoter regions T151C, A180G, and G318A, associated with these breeds and others, such as the Cocker Spaniel [40,43]. In our study, Cocker Spaniels had a high prevalence of infection (26.83%). The authors of [40,43] explained that these three SNPs of the *Slc11a1* gene are associated with a visceral presentation of the disease, which was similar in the different breeds analyzed. Although there are fewer studies on the presence of these haplotypes and SNPs in all dogs, the breeds with the lowest prevalence (Bichon, French Bulldog, Pug, and Chihuahua) and the highest prevalence (Doberman Pinscher, Boxer, Great Dane, Dogue de Bordeaux, and Boxer) appear to have common ancestors. A cladogram showed that their bootstrap supported more than 70% with over 150,000 SNPs [52,53]. Only the French Bulldog, genetically similar to the Boxer and Dogue de Bordeaux, presents a low seroprevalence of *L. infantum* infection in our study. Currently, there are no published scientific data to explain (in a genetic manner) the low prevalence of infection by *L. infantum* in this breed. 

To describe the infected population, although the cutaneous presentation is the most common in the dogs studied in this work regardless of breed and other factors, other authors found no relationship between sex and prevalence of *L. infantum* infection in Spain and Portugal [45,51]. Our results show similar seroprevalence in males and females, whereas some authors found a higher prevalence in males than females [54,55,56,57]. Viol et al., (2012) claimed these differences result from breeding males’ preference for surveillance or hunting activities, which exposes them to insect bites and infection by *L. infantum* [55]. Our data likely show similar seroprevalence because the dogs included in our study were not engaged in surveillance or hunting activities. The same authors [55] also found lower prevalence in adult dogs than younger and older dogs (>7 years), whereas Dantas-Torres et al., (2006) demonstrated a correlation between early age and prevalence [55,57]. Some studies indicated that the *L. infantum* infection presented two peaks in younger dogs (1–2 years) and a second peak at 7–8 years [58]. However, our results show no statistical difference between ages. According to studies performed in Brazil and Tunisia, the highest prevalence was found in dogs with clinical stage II, i.e., moderate disease [59,60].

We observed that antibody titers are associated with disease presentation; however, some authors ascribed the clinical picture to other factors such as eosinophil and alpha-globulin values [61]. We showed that animals with low antibody titers contracted the cutaneous form. These results are concordant with those obtained in a recent meta-analysis indicating that the cutaneous form could lead to more advanced stages of visceral disease [54]. The antibodies’ titer is related to clinical stage according to Martinez-Orellana et al., (2017) [49]. The authors argued that the clinical stage depends on the host’s immunologic response; therefore, the Th2 response produces ineffective antibodies and increases disease. Solano-Gallego et al., (2011) associated the clinical stage with antibody titers. Therefore, the clinical stage system is based on clinical signs, clinical–pathological abnormalities, and serologic status, directly associated with antibody titers of *L. infantum* [2].

## 5. Conclusions

Our study concludes that the total seroprevalence of *L. infantum* infection is 4.74% on a Mediterranean island, Ibiza Island. Our results confirm that the seroprevalence of *L. infantum* is linked to the canine breed but not with antibody titers. The differences in prevalence and breeds may be explained by three polymorphisms present in the *Slc11a1* gene. More studies should endeavor to document these interactions. Regarding the profile of seropositive animals, the distribution was similar to other studies. However, a relationship was observed between the number of antibody titers and the presentation of the disease. These data may indicate that the cutaneous presentation is anterior to the visceral one; however, future trials are needed. 

## Figures and Tables

**Figure 1 animals-11-02579-f001:**
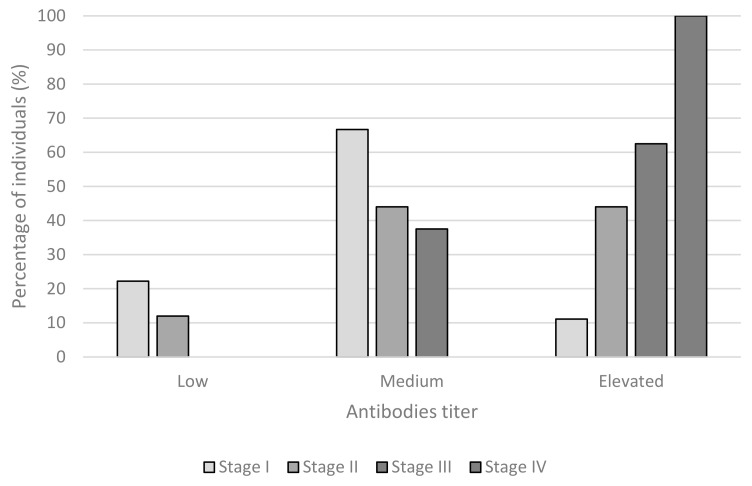
Relationship between clinical stage (stage I: mild disease, stage II: moderate disease, stage III: severe disease, and stage IV: very severe disease) and the presence of antibodies (low: <1:100; medium: between 1:100 and 1:400, and elevated: >1:400).

**Table 1 animals-11-02579-t001:** Classification of clinical stages based on clinical signs [15].

Clinical Stage	Clinical Signs
Stage I: mild disease	Mild clinical signs, peripheral lymphadenopathy, or popular dermatitis.
Stage II: moderate disease	Clinical signs of stage I include diffuse or symmetrical cutaneous lesions (exfoliative dermatitis, ulcerations), anorexia, weight loss, fever, and epistaxis.
Stage III: severe disease	Clinical signs of stages I and II, include vasculitis, arthritis, uveitis, and glomerulonephritis.
Stage IV: very severe disease	Clinical signs of stages I, II and III, include pulmonary thromboembolism, or nephrotic syndrome, and end-stage renal disease.

**Table 2 animals-11-02579-t002:** Prevalence of *Leishmania infantum* according to canine breed.

Breed	Studied Dogs	Seropositive Dogs	Seroprevalence (%)
Beagle	16	1	6.25 ^ab^
Maltese Bichon	68	1	1.47 ^a^
Boxer	23	9	39.13 ^c^
Bull Terrier	16	2	12.50 ^bc^
French Bulldog	426	6	1.41 ^a^
Pug	138	1	0.72 ^a^
Chihuahua	172	4	2.33 ^a^
Cocker Spaniel	41	11	26.83 ^c^
Doberman Pinscher	7	3	42.86 ^c^
Dogue de Bordeaux	3	1	33.33 ^c^
Golden Retriever	23	2	8.70 ^bc^
Great Dane	6	2	33.33 ^c^
Jack Russell	34	3	8.82 ^bc^
Labrador Retriever	103	14	13.59 ^bc^
Mastiff	22	2	9.09 ^bc^
German Shepherd	271	20	7.38 ^b^
Majorcan Shepherd	3	1	33.33 ^c^
Pinscher	55	2	3.64 ^ab^
Pitbull	18	1	5.56 ^ab^
Pointer	49	12	24.49 ^c^
Pomeranian	14	2	14.29 ^bc^
Schnauzer	8	1	12.50 ^bc^
Spitz	6	1	16.67 ^bc^
Weimaraner	3	1	33.33 ^c^
Yorkshire	47	3	6.38 ^ab^
Crossbreed	1569	43	2.74 ^a^
Overall	3141	149	4.74
	*p*-value		<0.05

^a, b, c^ Seroprevalence with no common superscripts differ significantly at *p* < 0.05.

**Table 3 animals-11-02579-t003:** Data recovered of seropositive dogs.

Variable Analyzed	Categorical Factors	Distribution of Seropositive (%)
Sex	Male	60.26
	Female	39.74
Age	Young (<4 years)	54.97
	Adult (4 ≤ years < 10)	38.41
	Older (≥10 years)	6.62
Presentation of disease	Cutaneous	62.91
	Visceral	35.10
	Both	1.99
Clinical stage	I	15.89
	II	58.28
	III	24.50
	IV	1.32

## Data Availability

Not applicable.

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
