# Peer review of "Is the Prevalence of Leishmania infantum Linked to Breeds in Dogs? Characterization of Seropositive Dogs in Ibiza"

_animals, 2021, doi:10.3390/ani11092579_

Round 1

Reviewer 1 Report

I read the manuscript titled “Is the prevalence of Leishmania infantum linked to genetic breeds in dogs? Characterization of seropositive dogs in Ibizan Island” and In my opinion the paper is not acceptable for its publication. Several major revision are needed before its publication.

It present several gaps in all sections. Particularly “material and methods” as well as “results” are the two sections that need more attention.

I would also suggest authors to check carefully the available literature, some example are reported below.  

The discussion is not clear and the conclusions, reported in “Discussion” section or in “Conclusion” section are not supported by the results presented.

Please ask to some native English speaker or a translation service to check the English. Several sentences need to be checked in the entire manuscript.

Specific changes:

Lines 41-42: “This zoonotic disease is endemic of 88 countries and is considered the most relevant vector-born disease after malaria [1,2]” This sentence mislead the reader, authors are speaking about Mediterranean regions or worldwide?

Line 43 “Leishmania genera” all Latin names has to be written in italic

Lines 47-48 “Host immunity … severe infection” this sentence needs a reference.

Lines 50-51: “whereas studies have not been carried out in Mediterranean islands” Please check carefully the literature: As an example, but not limited to these:  

10.1186/s13071-020-3992-8

10.1007/s00436-020-06973-0

10.12834/VetIt.2059.10976.3

10.1016/j.vetpar.2017.01.019

Particularly, please check this updated review: 10.1186/s13071-020-04081-7

Lines 52-53: Several publication from 2021 present studies about immune system of dogs infected with Leishmania. As example, but not limited to these:

10.3390/microorganisms9040712

10.1016/j.vetimm.2021.110198

10.1371/journal.pntd.0009137

10.1007/s00436-021-07091-1

10.1016/j.actatropica.2021.105906

10.3390/microorganisms9040712

10.1016/j.molimm.2021.06.014

10.1016/j.vetimm.2021.110196

10.1016/j.vetimm.2021.110198

10.1371/journal.pone.0239171

About genetic resistance in infected dogs, please also check:

10.1016/j.vetpar.2020.109276

10.1016/j.vetpar.2019.04.011

10.1016/j.ijpddr.2018.08.002

Lines 54-55 “However, some studies … pathological process.” this sentence needs a reference

Lines 62-70” The Th1 or Th2 activation does not depend only by the host, in this case the dog. It is the result of a host-pathogen interaction and it is more complex than what expressed by authors. Indeed, I would suggest to reformulate this paragraph.

Lines 71-72 “Some authors … in dogs” this sentence needs a reference

Lines 94-95 “… collected from September to May 2021 …” Please add the year for September.

Line 95 “…of 3,141 owner dogs” I believe this is an English mistake. What did the authors collected? Were all the dogs infected? How they determined it?

Lines 95-98 “The data recovered were prevalence … and clinical stages.” Please reformulate the sentence. It is confusing.

"2.2. Serologic tests": what kind of IFAT did the authors performed? An in-house or a commercial kit? If an in-house, please provide on which antigen it is based and briefly the methodology. If it is commercial, provide the full name of the kit and the manufacturer.

The results should be presented more clearly. I would suggest reviewing all tables, particularly table 4. Univariable and multivariable logistic analyses are not clearly presented

Table 2 is not clear, how it is possible that a dog's breed belongs to more than one group (first, second and third group, indicated as superscript a, b and c) ? Please indicate differently this aspect.

Lines 158-159: Please compare the prevalence you found with the prevalence of Baleares islands reported in literature (up to 57%) 10.1186/s13071-020-04081-7

Line 191: “However, our results showed higher seroprevalence in males than females, in … .”  There is no association between sex and seroprevalence. p-value of 0.46 is statistically not significant, meaning no association.

Lien 198 “Our results showed higher prevalence in dogs before ten years, which is … .” as for line 191 based on the statistical analysis (p-values 0.43) there is no association.

Based on the results, the authors can only conclude that they observed a higher seropositivity in one breed respect another but the relationship between the breed and the prevalence is not statistically significant (p-value 0.97), indeed based on their results there is no association.

The only association, statistically significant, observed in this study is between “presentation of the disease” and the “antibody titre”.

Line 203  “This is the first work where is observed that Antibodies titres are related with presentation of disease [45]” Is reference 45 placed in the correct position? Is this work, submitted to Animals journal, the first work where is observed that the antibody titres are related with the disease manifestation?

Line 204 “We showed that animals with low antibody titres had the cutaneous form” This is the first time that Authors are reporting that the analyzed dogs presented cutaneous form, how many dogs present cutaneous leishmaniasis, how many visceral? This aspect should be explain in "Material and Method".

Line 216 “Seroprevalence of L. infantum is linked with genetic breed.” This cannot be a conclusion of this study. Latin names should be written in italic.

I would suggest checking these articles

10.1371/journal.pone.0244923

10.1007/s00436-020-06973-0 

10.1016/j.prevetmed.2018.10.015

10.1590/S1984-29612012005000005

10.1186/s13071-020-3992-8

10.1016/j.vprsr.2017.12.006

10.1016/j.rvsc.2019.06.013

Author Response

I read the manuscript titled “Is the prevalence of Leishmania infantum linked to genetic breeds in dogs? Characterization of seropositive dogs in Ibizan Island” and, in my opinion the paper is not acceptable for its publication. Several major revisions are needed before its publication. It presents several gaps in all sections. Particularly “material and methods” as well as “results” are the two sections that need more attention. I would also suggest authors to check carefully the available literature, some examples are reported below. The discussion is not clear and the conclusions, reported in “Discussion” section or in “Conclusion” section are not supported by the results presented. Please ask to some native English speaker or a translation service to check the English. Several sentences need to be checked in the entire manuscript.

Regarding these comments, they are greatly appreciated. We are sure that they will greatly improve the manuscript. We have made several changes related to the sections that have been indicated to us previously. We have improved the materials and methods as well as the results, and established more data-driven conclusions. We have made an improvement in English (with a native speaker) and above all we would like to thank the bibliography provided.

Specific changes:

Lines 41-42: “This zoonotic disease is endemic of 88 countries and is considered the most relevant vector-born disease after malaria [1,2]” This sentence mislead the reader, authors are speaking about Mediterranean regions or worldwide?

Thank you for this comment. To clarify the sense of this sentence, it has been changed to “This zoonotic disease is endemic of 88 countries around the world and is considered the most relevant vector-born disease in Mediterranean area, where presents prevalence between 63 to 80% of the dog population” and three references that support these data have been added.

Line 43 “Leishmania genera” all Latin names has to be written in italic

Done.

Lines 47-48 “Host immunity … severe infection” this sentence needs a reference.

Done.

Lines 50-51: “whereas studies have not been carried out in Mediterranean islands” Please check carefully the literature: As an example, but not limited to these:

10.1186/s13071-020-3992-8

10.1007/s00436-020-06973-0

10.12834/VetIt.2059.10976.3

10.1016/j.vetpar.2017.01.019

Particularly, please check this updated review: 10.1186/s13071-020-04081-7

Thank you very much for this suggestion. The references were referring to Spanish Mediterranean islands. To improve understanding, the sentence has been changed to “Some studies have been carried out in Mediterranean islands on the prevalence of infection by L. infantum [18,19]. In the Spanish Mediterranean islands, Burnham et al. (2020) studied the relationship between susceptibility to canine leishmaniosis and anti-Phlebotomus pernicious saliva antibodies in Ibizan hounds, without analysing seroprevalence data in different dog breeds [20]. The global seroprevalence in Spanish Mediterranean islands was estimated around 57.1% [21].”, so the relevant references that the reviewer has indicated are studies carried out in Italian islands and the only study carried out in Spanish islands, does not indicate prevalence data in different canine breeds. However, we have added these references due to the importance of the information they provide in our manuscript.

Lines 52-53: Several publication from 2021 present studies about immune system of dogs infected with Leishmania. As example, but not limited to these:

10.3390/microorganisms9040712

10.1016/j.vetimm.2021.110198

10.1371/journal.pntd.0009137

10.1007/s00436-021-07091-1

10.1016/j.actatropica.2021.105906

10.1016/j.molimm.2021.06.014

10.1016/j.vetimm.2021.110196

10.1016/j.vetimm.2021.110198

10.1371/journal.pone.0239171

The information about immune system of dogs infected with L. infantum has been expanded, including some recent papers.

About genetic resistance in infected dogs, please also check:

10.1016/j.vetpar.2020.109276

10.1016/j.vetpar.2019.04.011

10.1016/j.ijpddr.2018.08.002

The information about genetic resistance in infected dogs and related with different breeds has been expanded, including some recent papers.

Lines 54-55 “However, some studies … pathological process.” this sentence needs a reference

We added two references in this sentence.

Lines 62-70” The Th1 or Th2 activation does not depend only by the host, in this case the dog. It is the result of a host-pathogen interaction and it is more complex than what expressed by authors. Indeed, I would suggest to reformulate this paragraph.

We fully agree with the reviewer's suggestion. We have reformulated the phrase in this way: The clinical pathogeny of L. infantum infection is a multifactorial process, where the activation of the Th1 or Th2 pathways depends on the interaction between host-pathogen interaction. The cellular immune response or Th1 pathway...”.

Lines 71-72 “Some authors … in dogs” this sentence needs a reference

This sentence has been deleted.

Lines 94-95 “… collected from September to May 2021 …” Please add the year for September.

Add 2020 year.

Line 95 “…of 3,141 owner dogs” I believe this is an English mistake. What did the authors collected? Were all the dogs infected? How they determined it?

Samples and breed data were collected of total dogs (3,141 dogs). The dogs with symptoms compatible with L. infantum infection underwent the serological test and data on sex, age, antibody titer, etc. were collected. To improve the understanding, the sentence has been changed to “. Data and samples were recovered from September 2020 to May 2021 of 3,141 owner dogs with symptoms compatible with L. infantum infection”.

Lines 95-98 “The data recovered were prevalence … and clinical stages.” Please reformulate the sentence. It is confusing.

The sentence has been reformulated as “Serological analysis was realized in all animals. In animals with positive serological test, data of breed (twenty-five levels), sex (two levels), age (divided into three categories: young [<4 years], adult [4 ≤ years <10] and older dogs [≥ 10 years]), presentation of disease (three levels: cutaneous, visceral, or both), and clinical stages were recovered.”

"2.2. Serologic tests": what kind of IFAT did the authors performed? An in-house or a commercial kit? If an in-house, please provide on which antigen it is based and briefly the methodology. If it is commercial, provide the full name of the kit and the manufacturer.

We used a commercial kit. The name of the kit and the manufacturer has been added.

The results should be presented more clearly. I would suggest reviewing all tables, particularly table 4. Univariable and multivariable logistic analyses are not clearly presented.

The table 4 has been eliminate, because we have in according with the reviewer, and we considered that table 4 does not provide relevant extra information that cannot be explained in the text.

Table 2 is not clear, how it is possible that a dog's breed belongs to more than one group (first, second and third group, indicated as superscript a, b and c)? Please indicate differently this aspect.

The binary logistic regression model with a genetic breed as fixed effect indicates statistically statistical similarities and differences in the prevalence of infection according to breed. In this way, the animals that present more than one superscript means that they do not present statistically significant differences. For example, Beagle breed dogs (with superscripts a and b) show statistically significant differences only with boxer (superscript c), and are similar (in prevalence data) to the Maltese Bichon (superscript a) and also to the German Shepherd.

Lines 158-159: Please compare the prevalence you found with the prevalence of Baleares islands reported in literature (up to 57%) 10.1186/s13071-020-04081-7

The prevalence obtained in our work has been compared with others published, not only in the work indicated by the reviewer, but in others. In order to add this interesting fact to our manuscript, the following paragraph has been added in the discussion “Concretely, in Balearic Islands, the prevalence has been estimated between 1% in 813 dogs examined in the year 1999 [46] to 40% during 2011-2016 in 40 dogs examined [21]. These relevant differences could be due to the sample size used. To date, our work is the first in which more than 3000 dogs on the island of Ibiza are analyzed”.

Line 191: “However, our results showed higher seroprevalence in males than females, in … .” There is no association between sex and seroprevalence. p-value of 0.46 is statistically not significant, meaning no association.

Thank you very much for your comment. We have in agree with that, and the paragraph has been changed as “However, our results show similar seroprevalence in males than females, whereas some authors founded higher prevalence in males than females [53–55]. Viol et al. (2012) explain these differences with the preference for breeding males for surveillance or hunting activities, which exposes them more to insect bites and, therefore, to infection by L. infantum [53]. Probably, our data showed similar seroprevalence because the animals included in our study were not dogs engaged in surveillance or hunting activities”.

Line 198 “Our results showed higher prevalence in dogs before ten years, which is … .” as for line 191 based on the statistical analysis (p-values 0.43) there is no association.

Thank you very much for your comment. We have eliminated this sentence of the discussion.

Based on the results, the authors can only conclude that they observed a higher seropositivity in one breed respect another but the relationship between the breed and the prevalence is not statistically significant (p-value 0.97), indeed based on their results there is no association. The only association, statistically significant, observed in this study is between “presentation of the disease” and the “antibody titre”.

We observed differences between breeds in seropositive prevalence. P-value = 0.97 is referred to relationship between antibodies titre and breed in infected animals (animals with IgG>1/100). In results section, we have indicated the sentence: “Non effects were showed between antibodies titres with canine breed, sex or age.”

Line 203 “This is the first work where is observed that Antibodies titres are related with presentation of disease [45]” Is reference 45 placed in the correct position? Is this work, submitted to Animals journal, the first work where is observed that the antibody titres are related with the disease manifestation?

It is correct. The sentence has been improved as “This is the first work where is observed that Antibodies titres are related with presentation of disease, although some authors have related the clinical picture to other factors such as eosinophil and alpha-globulin values [45]”.

Line 204 “We showed that animals with low antibody titres had the cutaneous form” This is the first time that Authors are reporting that the analyzed dogs presented cutaneous form, how many dogs present cutaneous leishmaniasis, how many visceral? This aspect should be explained in "Material and Method".

In the material and methods section, concretely in subsection “Data and collection of samples” we have indicated that: “data of breed (twenty-five levels), sex (two levels), age (divided into three categories: young [<4 years], adult [4 ≤ years <10] and older dogs [≥ 10 years]), presentation of disease (three levels: cutaneous, visceral, or both), and clinical stages were recovered”

Line 216 “Seroprevalence of L. infantum is linked with genetic breed.” This cannot be a conclusion of this study. Latin names should be written in italic.

The seroprevalence presents relationship statistically demonstrated with the canine breed, but not with the antibody titre. We have changed the conclusion as “Seroprevalence of L. infantum is linked with canine breed, but not with the antibody titre”.

I would suggest checking these articles

10.1371/journal.pone.0244923

10.1007/s00436-020-06973-0 

10.1016/j.prevetmed.2018.10.015

10.1590/S1984-29612012005000005

10.1186/s13071-020-3992-8

10.1016/j.vprsr.2017.12.006

10.1016/j.rvsc.2019.06.013

Thank you very much for your suggestion. We have revised these references and we have added some of them.

Reviewer 2 Report

See file Review.

Author Response

We already answered the reviewer in a previous review. However, we will resend the responses to your comments.

Reviewer 3 Report

The authors investigated the Leishmania sp. infection by IFAT in dogs from Ibizan Island, in Spain, aiming to describe any relationship with the genetic breed. The proposal is very interesting and grounded in similar recent articles that suggest the genetic background as an important aspect to be considered in canine leishmaniasis. However, I identified some aspects that need to be considered. The major points are:

1) Nothing was stated about the ethical content. This manuscript includes dogs’ manipulation and sample collection and should had been approved by an Ethical Committee before its realization.

2) Some variables not related to the infection (breed, sex and age) were also collected from non-infected dogs and should be presented here. Comparisons should have been done between groups of infected and non-infected dogs. Example: Sex: distribution of seropositive was 60/40 (males/females). But what was the sex distribution among all dog population (n=3141) from Island? Is your result reflecting that more males are infected or just reflecting the normal distribution of males and females in the whole population? The same approach should be done for all the variables. This was considered only for breed, but not for sex and age. Statistical analysis is important to be done.

3) Table 1: Were laboratory tests performed on any of dogs? It is not listed in the methodology. If not, delete the last column from table 1.

4) In the absence of other confirmatory tests (parasitological and/or molecular), authors should have used two different serological tests (IFAT and ELISA, for example) as preconized for serological diagnosis in any infectious disease. Defining infected and uninfected animals based only on one unique serological test is a very error-prone methodology. A confirmatory test has to be employed in all samples.

5) Lines 109-111: If the adopted cutoff was 1:100, how values less than 1:100 are listed in the ranking of positives (low or questionable)? Samples with values lower than 1:100 should be considered only as negative.

6) Considering the two breeds described by authors as the most infected ones (Doberman Pinscher and Boxer), the number of dogs examined in these two breeds was very small (23 Boxer and 7 Doberman Pinscher) – Table 2. Are any of them from the same tutors or geographic origin? This information is crucial because they may have the same common source of infection. Even more so in such a low amount, this must be considered.

7) French Bulldog was the most investigated breed (426 animals), far from others. Could the low infection rate observed be a diluting effect of this large amount of animals analyzed (which was not observed in any other breed)? In fact, apart from occasional exceptions, the highest infection rates were observed in breeds where fewer than 40 animals were analyzed (Table 2). This should be discussed by authors.

8) Authors pointed that “Antibodies titres was correlated with presentation of disease” and that “all animals with low antibodies titres showed a cutaneous presentation”(lines 141-143). However, most of dogs that presented moderate and high titres also showed a cutaneous presentation. In fact, the cutaneous presentation represented 38/48 positives (Table 4).

9) Table 4: There are 149 positive dogs, but the sum of the categories (breeds, sex, age or clinical presentation) gives much smaller numbers. E.g. adding males and females, in the 3 categories of intensity of serological titer we have 43 animals only.

MINOR POINTS:

ABSTRACT:

Revise the English spelling in the second sentence;

Lines 26-27: genetic breed, age and sex were parameters evaluated for all examined dogs, infected or dog, right? Rephrase this sentence in a clearer manner;

Lines 32-33: It was not observed a direct relationship between the antibodies titters with the presentation of the disease. This relationship was observed only for one aspect, as pointed in lines 142-143: “all animals with low antibodies titres showed a cutaneous presentation; p-value < 0.05”.  Correct it;

Lines 34-35: The last sentence is an observation from another study, which was not discussed in depth in this one. This should not be included in the article abstract;

INTRODUCTION:

Line 40: L. infantum in parenthesis is not necessary for scientific names

MATERIAL AND METHODS

Line 95: September 2020 to May 2021, isn't it? Was this manuscript submitted only 2 months after the sample collection?

DISCUSSION:

Are there any study regarding vector distribution on Ibiza Island? It should be included and discussed in the article.

Lines 187-189: Revise the English spelling

Lines 191-192 and 198-199: Comparisons should include all population (infected and non-infected) and demonstrated with statistical analysis.

CONCLUSION:

Lines 271-218: This is not a conclusion of this work, but a discussion from other studies. This should not be included in the conclusion.

Author Response

The authors investigated the Leishmania sp. infection by IFAT in dogs from Ibizan Island, in Spain, aiming to describe any relationship with the genetic breed. The proposal is very interesting and grounded in similar recent articles that suggest the genetic background as an important aspect to be considered in canine leishmaniasis. However, I identified some aspects that need to be considered. The major points are:

Thank you very much for all suggestions. We are sure that they will greatly improve the manuscript. Below, we have responded to each of the suggested points:

1) Nothing was stated about the ethical content. This manuscript includes dogs’ manipulation and sample collection and should had been approved by an Ethical Committee before its realization.

Sorry for this mistake. The information about approval by the Ethical Committee of our institution has been added.

2) Some variables not related to the infection (breed, sex and age) were also collected from non-infected dogs and should be presented here. Comparisons should have been done between groups of infected and non-infected dogs. Example: Sex: distribution of seropositive was 60/40 (males/females). But what was the sex distribution among all dog population (n=3141) from Island? Is your result reflecting that more males are infected or just reflecting the normal distribution of males and females in the whole population? The same approach should be done for all the variables. This was considered only for breed, but not for sex and age. Statistical analysis is important to be done.

We know that it would be important to have this data. However, we do not have them, since the objective of the work was, mainly, to analyze the prevalence according to the dog breed and the characterization of seropositive dogs. For this reason, the only data collected from all the animals (3,141) was breed. This is the explanation why a statistical analysis like the one you mentioned has not been carried out and we have limited ourselves to making a description of the seropositive population.

3) Table 1: Were laboratory tests performed on any of dogs? It is not listed in the methodology. If not, delete the last column from table 1.

The last column from table 1 has been deleted.

4) In the absence of other confirmatory tests (parasitological and/or molecular), authors should have used two different serological tests (IFAT and ELISA, for example) as preconized for serological diagnosis in any infectious disease. Defining infected and uninfected animals based only on one unique serological test is a very error-prone methodology. A confirmatory test has to be employed in all samples.

We know the limitations indicated by the reviewer. However, we currently do not have the samples for further serological analysis. However, some studies indicate that the IFAT test is sufficient to detect seropositivity in dogs with a reliability greater than 95%, similar to other serological tests (Chargui et al., 2009). Obviously, molecular tests are much more specific, but IFAT tests are tests of choice in the current veterinary clinic, so we decided to use this test for our analyses, as in other previously published studies (Spada et al., 2016; Clemente et al., 2014; Ferreira et al., 2013; Adel et al., 2015; among others).

5) Lines 109-111: If the adopted cutoff was 1:100, how values less than 1:100 are listed in the ranking of positives (low or questionable)? Samples with values lower than 1:100 should be considered only as negative.

Sorry for our mistake. The cut-off was 1:80. The number has been changed in the manuscript (material and methods section).

6) Considering the two breeds described by authors as the most infected ones (Doberman Pinscher and Boxer), the number of dogs examined in these two breeds was very small (23 Boxer and 7 Doberman Pinscher) – Table 2. Are any of them from the same tutors or geographic origin? This information is crucial because they may have the same common source of infection. Even more so in such a low amount, this must be considered.

The authors took into account this possible bias indicated by the reviewer. Neither of the animals of these two breeds came from the same tutors or dwellings. Although the sample size is low in these two breeds, compared to others analyzed, the statistical analysis carried out ensures a statistically significant difference, even taking into account this different sample size between breeds.

7) French Bulldog was the most investigated breed (426 animals), far from others. Could the low infection rate observed be a diluting effect of this large amount of animals analyzed (which was not observed in any other breed)? In fact, apart from occasional exceptions, the highest infection rates were observed in breeds where fewer than 40 animals were analyzed (Table 2). This should be discussed by authors.

The reviewer is correct that the sample sizes of the different breeds analyzed are very diverse. However, we believe that the robust statistical analysis carried out allows us to draw the conclusions of the work, despite this limitation. On the other hand, this limitation is due to the characteristics of the study, where data were collected from veterinary clinics on the island of Ibiza, which has a small geographic size of 571.6 km². In addition, other studies related to differences between canine breeds have used similar or smaller sample sizes, for example, the study of Parker et al. (2004) in the scientific journal Science, where the authors used a total of 152 dogs of 28 breeds to detect specific genetic marker to breed, and another (Burnham et al., 2020; Martínez-Orellana et al., 2017; Sanchez-Roberts et al., 2005).

8) Authors pointed that “Antibodies titres was correlated with presentation of disease” and that “all animals with low antibodies titres showed a cutaneous presentation”(lines 141-143). However, most of dogs that presented moderate and high titres also showed a cutaneous presentation. In fact, the cutaneous presentation represented 38/48 positives (Table 4).

Thank you very much for your comment. The sentence “all animals with low antibodies titres showed a cutaneous presentation” is a mistake. This sentence has been deleted.

9) Table 4: There are 149 positive dogs, but the sum of the categories (breeds, sex, age or clinical presentation) gives much smaller numbers. E.g. adding males and females, in the 3 categories of intensity of serological titer we have 43 animals only.

Again this is a mistake. To avoid confusion, and since table 4 did not provide additional information to the text, we have eliminated table 4.

MINOR POINTS:

ABSTRACT:

Revise the English spelling in the second sentence;

Revised and corrected.

Lines 26-27: genetic breed, age and sex were parameters evaluated for all examined dogs, infected or dog, right? Rephrase this sentence in a clearer manner;

The sentence was rewritten as “For to study the prevalence, a total of 3,141 dog were analyzed and the genetic breed was recovered. In the infected animals (n=149) was studied age, sex, antibody titre, and stage of the disease”.

Lines 32-33: It was not observed a direct relationship between the antibodies titters with the presentation of the disease. This relationship was observed only for one aspect, as pointed in lines 142-143: “all animals with low antibodies titres showed a cutaneous presentation; p-value < 0.05”.  Correct it;

The sentence was correct as “Finally, regarding the characterization of seropositive animals, the observed distribution was similar than other studies but it has a relationship (p < 0.05) between the number of antibodies titters with the clinical stage of disease.

Lines 34-35: The last sentence is an observation from another study, which was not discussed in depth in this one. This should not be included in the article abstract;

The last sentence has been deleted.

INTRODUCTION:

Line 40: L. infantum in parenthesis is not necessary for scientific names

Deleted.

MATERIAL AND METHODS

Line 95: September 2020 to May 2021, isn't it? Was this manuscript submitted only 2 months after the sample collection?

The year of September 2020 was added. The sample collection was realized between September 2020 to May 2021.

DISCUSSION:

Are there any study regarding vector distribution on Ibiza Island? It should be included and discussed in the article.

Vector distribution on Ibizan Island and sand fly vector founded was added in discussion (lines 198-201).

Lines 187-189: Revise the English spelling

Done and corrected.

Lines 191-192 and 198-199: Comparisons should include all population (infected and non-infected) and demonstrated with statistical analysis.

Data recovered of canine breed has been statistically analyzed. The other data (sex, age, etc) were recovered only in infected animals. For this reason, the statistical analysis has not been carried out and we have indicated a descriptive analysis of infected animals.

CONCLUSION:

Lines 271-218: This is not a conclusion of this work, but a discussion from other studies. This should not be included in the conclusion.

The sentence has been removed to discussion section.

Round 2

Reviewer 1 Report

I check the reviewed manuscript and I notice the  improvement made by authors following the suggestions. I think the English were at the basis of misunderstanding between what the authors wanted to say and what actually they were saying. Unfortunately, despite they changed the pointed sentences, this aspect is still present.

Two main points need to be addressed:

1 - Methods are still not clear, and they should be clarify. IFAT analysis is the most wondering for me, I would suggest to clarify it otherwise authors would need to analyze again the raw data.  Lines 137-138 “Dog serum were considered seropositive with IFAT titre ≥ 1:100 [44,45].” Why authors are using a cut-off of 1:100? Did Mirò et al, and Olias-Molero et al, the two cited article, use the MegaFluo Leish® kit? What is the cut-off indicated in the manufacturer instruction? If authors did not follow the manufacturer instruction, why they did not?  

Are the raw data available within the manuscript or they have been deposited in some repository?  Please provide them within the submission.

2 -Lines 119-120 “Data and samples were recovered from September 2020 to May 2021 of 3,141 owner dogs with symptoms compatible with L. infantum infection …” What kind of samples did they collected? Did authors performed the study on the dogs or on the human owner of the dogs? As I already suggested, please check the English. I strongly believe they took samples from the dogs and not from the owner of the dogs, but this is not what is written in there.  What were the symptoms compatible with L. infantum infection? Lines 126 and 127 should be linked to this sentence. How much dogs were symptomatic? With cutaneous and/or visceral? Among the different breed how much presented symptoms? How much of symptomatic dogs were seropositive? How much negative? I would suggest to add a table with all these information.  Were the symptoms specific of Leishmaniasis or general symptoms? How do they correlate the symptoms with Leishmaniosis?

This reviewed version is not suitable for publication, major changes are still needed.  Please find below specific suggestions that can help improving your manuscript:

It is not clear to me how “ … this work suggests a very important genetic component ... ” ?

 “ Finally, regarding the characterization of seropositive animals, the observed distribution was similar than other studies but it has a relationship (p < 0.05) between the number of antibodies titters with the clinical stage of disease. “Unfortunately this do not represent such a novelty

Line 43 “ This zoonotic disease is endemic of 88 countries around the world ... “ Leishmaniasis can be zoonotic, anthropozoonotic and also anthroponotic. Leishmaniasis caused by L. infantum is a zoonotic disease, this is correct. Worldwide Leishmaniasis are not only zoonotic diseases. I would suggest to ask to some parassitologist to check this part.

Lines 50-51 “ Host immunity response determines the infection severity of L. infantum, can be a mild, moderate, severe or very severe infection …” Since Leishmaniasis are a group of diseases that can be cutaneous, mucocutaneous or visceral maybe authors should indicate what symptoms an infection of L. infantum in dogs can produce.

Line 76 “…the arginase enzyme, essential for the nutrition of the parasite…” L-arginine is an essential amino acid for Leishmania spp. The protozoa possess arginase, I would suggest to read this article 10.1016/j.molbiopara.2011.10.007 for better understand the biochemistry of Leishmania spp.

Line 87: “Different prevalence between dog breeds …” What prevalence?

Line 174: “ … correlated with presentation of disease, but not with canine breed, sex, and age ..” Did authors performed a correlation analysis? Their results do not present an association between breed, sex, age and seroprevalence.

Line 247: “This is the first work where is observed that Antibodies titres are related with presentation of disease” Please check this study 10.1155/2014/412808 Moreover, line 252-258 they are affirming the contrary.

Table 2 is still not clear. As Authors reply : “The binary logistic regression model with a genetic breed as fixed effect indicates statistically statistical similarities and differences in the prevalence of infection according to breed. In this way, the animals that present more than one superscript means that they do not present statistically significant differences. For example, Beagle breed dogs (with superscripts a and b) show statistically significant differences only with boxer (superscript c), and are similar (in prevalence data) to the Maltese Bichon (superscript a) and also to the German Shepherd.”

Unfortunately the statistical analysis they are explaining to me are not reported in the text. Please explain the results and the table in the text also.

Is it possible that the different sample size of each group of dog’s breed influenced these results? I would suggest to the authors also to explain this in the text.

Author Response

(The authors gave the same response as above.)

Reviewer 3 Report

The authors answered all my questions; however, some limitations of the article remain and could not be resolved due to lack of information and lack of material for further analysis. Four points in particular represent important limitations that authors cannot elucidate:

1) Information on parameters that are analyzed by the authors, such as sex and age, were not collected for non-infected animals, which makes comparisons between infected and non-infected impossible. Without this comparison, between the infected group and the total population, it is not possible to carry out the analysis as it would be necessary to obtain the conclusions presented by the authors.

2) Although authors report that IFAT is a reliable serological test, as for any serological test, the higher is the sensibility, the lower is the specificity. For this reason, in the absence of parasitological and other (necessary) serological tests, the accuracy of the diagnosis is impacted. It is known that other hemoparasites, as Babesia and Erlichia, may cross react and result in false positive diagnosis.

3) In one of the answers, authors inform that “data were collected from veterinary clinics”, which was up to now not clear to me. This provides an additional bias in the study since the descriptions of clinical presentation (an attribute that may vary in different observer) were performed by different vet clinicians.

4) Other information added in the present version, is that only dogs “with symptoms compatible with L. infantum infection” were included. And this also represents a bias, because asymptomatic dogs were not included in the analysis. It is know that some infected dogs do not display symptoms and this may be related to a successful immune control of the parasitosis. Maybe some breeds are more competent to do this than other, and these asymptomatic were not included in the manuscript.

Author Response

(The authors gave the same response as above.)
